# Redox Reactivity of Nonsymbiotic Phytoglobins towards Nitrite

**DOI:** 10.3390/molecules29061200

**Published:** 2024-03-07

**Authors:** Cezara Zagrean-Tuza, Galaba Pato, Grigore Damian, Radu Silaghi-Dumitrescu, Augustin C. Mot

**Affiliations:** 1Department of Chemistry, Faculty of Chemistry and Chemical Engineering, Babes-Bolyai University, Arany Janos Str. No. 11, RO-400028 Cluj-Napoca, Romania; cezara.zagrean@ubbcluj.ro (C.Z.-T.); gabi_naumova@hotmail.com (G.P.); radu.silaghi@ubbcluj.ro (R.S.-D.); 2Faculty of Physics, Babes-Bolyai University, Mihail Kogalniceanu Str. No. 1, RO-400084 Cluj-Napoca, Romania; grigore.damian@ubbcluj.ro

**Keywords:** hemoglobin, nitrite, phytoglobins, redox reactivity, nonsymbiotic hemoglobin, nitric oxide, nitrite reductase activity, nitrite oxidase activity

## Abstract

Nonsymbiotic phytoglobins (nsHbs) are a diverse superfamily of hemoproteins grouped into three different classes (1, 2, and 3) based on their sequences. Class 1 Hb are expressed under hypoxia, osmotic stress, and/or nitric oxide exposure, while class 2 Hb are induced by cold stress and cytokinins. Both are mainly six-coordinated. The deoxygenated forms of the class 1 and 2 nsHbs from *A. thaliana* (AtHb1 and AtHb2) are able to reduce nitrite to nitric oxide *via* a mechanism analogous to other known globins. NsHbs provide a viable pH-dependent pathway for NO generation during severe hypoxia *via* nitrite reductase-like activity with higher rate constants compared to mammalian globins. These high kinetic parameters, along with the relatively high concentrations of nitrite present during hypoxia, suggest that plant hemoglobins could indeed serve as anaerobic nitrite reductases *in vivo*. The third class of nsHb, also known as truncated hemoglobins, have a compact 2/2 structure and are pentacoordinated, and their exact physiological role remains mostly unknown. To date, no reports are available on the nitrite reductase activity of the truncated AtHb3. In the present work, three representative nsHbs of the plant model *Arabidopsis thaliana* are presented, and their nitrite reductase-like activity and involvement in nitrosative stress is discussed. The reaction kinetics and mechanism of nitrite reduction by nsHbs (deoxy and oxy form) at different pHs were studied by means of UV-Vis spectrophotometry, along with EPR spectroscopy. The reduction of nitrite requires an electron supply, and it is favored in acidic conditions. This reaction is critically affected by molecular oxygen, since oxyAtHb will catalyze nitric oxide deoxygenation. The process displays unique autocatalytic kinetics with metAtHb and nitrate as end-products for AtHb1 and AtHb2 but not for the truncated one, in contrast with mammalian globins.

## 1. Introduction

A role for nitric oxide in plant physiology was first described in 1998 as a factor in plant disease resistance [1]. Since then, a plethora of other functions have been uncovered, as nitric oxide is now a recognized essential signaling player in all stages of plant life, with roles in seed germination, root development, photosynthesis regulation, senescence, and root nodule formation in legumes, to name a few [2,3,4,5,6,7,8,9,10,11]. While nitric oxide synthesis in mammals was first described in the late 1980s [12], its synthesis in plants remains elusive due to the lack of a typical nitric oxide synthase (NOS) [13]. Possible biosynthetic pathways may include NOS-like activities of nitrate reductase (NR), nitrite reductase (NiR), and other heme-containing enzymes found in various cell organelles [14,15,16,17,18,19,20]. Generally speaking, nitrate reductase or NR-like enzymes are able to generate nitrite in the presence of NADPH as an electron donor; subsequently, nitrite is converted to nitric oxide by nitrite reductase or NiR-like enzymes using soluble cytochrome c or blue copper protein as electron donors [21]. 

Hemoglobins (Hbs) are a diverse family of proteins with numerous biological functions in all kingdoms of life [22], including plants, where symbiotic and nonsymbiotic phytoglobins are characterized; the latter type comprises three classes: class 1, class 2, and class 3 (truncated hemoglobins—trHb) [23,24,25]. They are expressed during stress scenarios, but their specific function remains elusive [26,27]. Class 1 nonsymbiotic phytoglobins (Hb1) are the best understood so far, and their expression occurs under hypoxia, cold stress, osmotic stress, and fungal infection, now recognized as part of a NO dioxygenase system [28,29,30,31,32]. Class 2 nonsymbiotic phytoglobins are induced by low temperatures (cold stress) and drought stress, and their functions are most likely related to oxygen storage, diffusion, and supplying oxygen to developing tissues [33,34,35]. Truncated phytoglobins (class 3) are even more obscure in their evolution and function *in planta*; the lack of concentration-dependent binding of both O_2_ and CO makes them even more puzzling [36]. 

In the early 2000s, mammalian hemoglobin joined the NiR-like enzyme group, as the conversion of nitrite to nitric oxide was observed under hypoxic conditions, contributing to vasodilation [37,38,39]. Over the next decade, in vitro NiR-like activity was observed in nonsymbiotic class 1 hemoglobins of both *Oryza sativa* and *Arabidopsis thaliana* [18,40]. While an increase in the gene expression of class 1 phytoglobins is mostly associated with hypoxic treatment [41,42,43,44], interesting regulation patterns were observed for both class 2 (down-regulation in anoxic conditions in *A. thaliana*) [45] and class 3 truncated (up-regulation in hypoxic grey poplars, followed by down-regulation after 24 h) phytoglobins [46].

A thorough a kinetic exploration of *A. thalina* phytoglobins’ (met, oxy, and deoxy) reactivity towards nitrite is reported herein, in comparison to mammalian myoglobins, all deoxyphytoglobins exhibit nitrite reductase-like activity to different extents. Class 1 phytoglobins are characterized by having the best kinetic parameters, while truncated class 3 phytoglobins display NiR-like reactivity slightly better than that of myoglobins. Oxyphytoglobins are able to convert nitrite to nitrate; it is worth noting that oxyAtHb3 especially has a very high rate for nitrite conversion, most probably due to the high reactivity of its ferryl intermediate. These high-rate constants suggest that phytoglobins, especially deoxyAtHb1, could indeed serve as anaerobic nitrite reductases *in vivo*.

## 2. Results and Discussion

### 2.1. Structural Particularities and Ferric Phytoglobins Binding Nitrite 

Although the nonsymbiotic phytoglobins are frequently classified and discussed based on their phylogenetics, oxygen affinity, physiological role, and structural aspects, such as 2/2 or 3/3 α-helical folding, another classification that is usually convenient in spectroscopical studies is based on iron coordination in heme while in deoxygenated form [47]. As illustrated in Figure 1, in the case of the hexacoordinated Hbs such as the AtHb1 (class 1) and AtHb2 (class 2) nonsymbiotic phytoglobins, the highly conserved and so-called distal histidine directly binds to the iron from heme with different strengths depending on the phytoglobins class. On the other hand, truncated phytoglobins such as the AtHb3 (class 3) discussed in this work exhibit a more compact 2/2 helical fold, homologous to bacterial hemoglobins; they lack the distal histidine and have a so-called pentacoordinated heme, along with particular spectral features in UV-vis spectra [25,48,49]. Moreover, the presence of both tyrosine and tryptophan in the binding pocket of these truncated hemoglobins may entail interesting redox properties. In the case of mammalian globins, this key distal histidine is not bound to the iron but is instead involved in stabilizing the iron-coordinated molecular oxygen [50]. Since the coordination of the distal histidine proved to play a crucial role in phytoglobin reactivity towards hydroxylamine [51], it is considered important to discuss the results considering this characteristic.

Even though ferric phytoglobins, or the so-called met form, presumably constitute a lower proportion of the entire Hb population *in planta*, due to the reducing environment, we consider them to be important from a mechanistic point of view. Therefore, we first tested the binding ability of very stable met nsHbs towards nitrite using UV-vis spectroscopy (Appendix A). While in the cases of metAtHb1 and metAtHb2, binding of the nitrite could not be detected even up to tens of millimolar range, due to the coordination of the distal histidine, ferric AtHb3 bound nitrite with a dissociation constant of 1.5 ± 0.5 mM, as shown in Figure 2. A bathochromic shift from 409 nm to 417 nm was seen in the Soret band, and a change from the typical aqua ferric form to the distinct low-spin shape was easily observed in the 480–680 nm spectral range. Although the binding mode of nitrite to iron in different proteins is frequently presented in various isomeric linkage forms, in the case of hemoproteins, N-nitrito and O-nitrito binding modes are still the most accepted ones. The ferric myoglobin binding of nitrite can be well characterized spectroscopically and crystallographically [52,53]. The determined K_D_ value of metAtHb3 is one order of magnitude lower than the wild-type myoglobin and very close to the H64Q mutant [53], indicating that the environment of the binding pocket, especially the distal histidine, plays an essential role in nitrite binding. A very interesting reversible thermally induced spin crossover in the myoglobin–nitrito adduct has also been recently observed, as revealed by Raman spectroscopy [54].

### 2.2. Ferrous Phytoglobins Nitrite Reductase Activity in Anaerobic Conditions 

In anaerobic conditions, mammalian myoglobin and hemoglobin are known to act as nitrite reductases, converting nitrite to nitric oxide, thus modulating NO signaling in mitochondrial respiration and other responses to cellular hypoxia and reoxygenation [39,55]. In the case of plants, an increasing amount of data indicate that both nitric oxide scavenging and nitrite reduction are highly important when it comes to regulating tolerance to hypoxia [7]. Although the detection of an HbFe^2+^-NO_2_^−^ adduct is only possible *via* photoreduction using X-rays [56], the nitrite reductase activity can be easily investigated using UV-vis spectroscopy. Figure 3 presents the detailed findings of this activity for the studied phytoglobins. A shift in the Soret band and the formation of the nitrosyl feature in the 475–600 nm spectral domain, with clear isosbestic points, is present in all three phytoglobins and myoglobin. These findings support previous observations that indicate that nitrite reduction leads to nitric oxide, while ferric Hb is reduced by dithionite [57]. In this case, dithionite is used as an electron source, replacing the corresponding plant methemoglobin reductase, which would be responsible for the conversion of met in deoxy form *in vivo* [58]. The kinetic curves could be easily fitted with a simple A → B process, as the reactions were performed in the presence of dithionite, and deoxy was the only phytoglobin form present. The normalized kinetic profiles and the rate constant are shown in Figure 4, whereas Table 1 lists the rate constants for the bimolecular reaction, considering the pseudo first-order conditions of the employed experiment. Given the use of a pH 7 buffer, the magnitude of the rate constant of both deoxyAtHb1 and deoxyMb are in good agreement with those determined in a previous work [40]; the rate constant for the bimolecular reaction for AtHb2 is about three times higher than that previously reported. 

To the best of our knowledge, this is the first case where the truncated Hb from *A. thaliana* has been shown to exhibit nitrite reductase activity. However, it exhibits a lower reactivity compared to that of hexacoordinated phytoglobins from the same organism, but its reactivity is still higher than the mammalian myoglobin.

### 2.3. Oxy Phytoglobins Nitrite Oxidase Activity in Aerobic Conditions

We have shown in our previous works that in the presence of oxygen or hydrogen peroxide, mammalian globins and phytoglobins have particularly striking differences, including reactivity towards hydrogen sulfide and interactions with plant phenolics [59,60,61,62]. The very high molecular oxygen affinity of phytoglobins, especially in classes 1 and 2, supports the idea that even in very drastic hypoxic conditions, phytoglobins will be oxygenated. On the other hand, although not directly involved with phytoglobins, nitrite was previously shown to be linked to hypoxic stress conditions [7]. Considering all this, a putative nitrite oxy phytoglobins interaction *in planta* cannot be ruled out, and thus, such oxyphytoglobin reactivity towards nitrite should be investigated. To the best of our knowledge, such data are not available yet. In the case of mammalian hemoglobin and myoglobin, the mechanism is known to be complex [60,61,62]. In these studies, it is accepted that the mechanism of this reaction involves an initial slow phase (lag phase) that accelerates into a rapid phase (propagation phase) of oxidation [62]. The lag time in minutes for this slow phase and the maximum decay rate in the fast phase were experimentally determined for all three phytoglobins and myoglobin (Figure 5 and Appendix A). The reaction profile was further investigated at different pH values. The reaction of oxyAtHb1 with nitrite showed a much longer lag time—between 10 and 60 min—compared to that of AtHb2—between 0.5 and 6 min—as shown in Figure 5 and Appendix A. Moreover, a pH increment as small as 0.05 caused a remarkable difference in the lag time of oxyAtHb1, which was not the case for oxyAtHb2. OxyAtHb3 exhibited a different kinetic behavior, with the presence of only the rapid phase or propagation phase in all of the conditions investigated, as presented in Appendix A. Since high-valent ferryl globin was detected as intermediate before [63,64,65], it is reasonable to assume that AtHb3 exhibits a totally different behavior in the presence of peroxide due to the formation of a very reactive Fe(IV) = O. 

As could be easily expected, for the three globins that exhibit the initial slow phase, the lag period decreases as the nitrite concentration increases; however, this decrease is nonlinear and approaches zero at concentrations of around 2 mM for AtHb1, about 0.5 mM for AtHb2, and 0.2 mM for myoglobin. At concentrations higher than these values, the reactions practically go directly to the propagation phase. On the other hand, the maximum decay rate—the rate at the inflection point of the kinetic profile—increases quasilinearly as the nitrite concentration increases. The rates of the reaction were favored in acidic pH, by both reducing the lag phase and increasing the maximum decay rate. This can be easily explained by the fact that the ferryl intermediate that is generated during the reaction has a much higher reactivity at acidic pH [66] and thus leads to a faster oxidation of the nitrite. At pH values below 6.0–6.5, the reaction does not exhibit a detectable lag phase.

For these in vitro studies, it must be mentioned that autoxidation rates should be taken into consideration when analyzing the reactions of oxyHbs with nitrite. Therefore, the long lag phase for AtHb1 and very short ones for AtHb2 and AtHb3 could also be partly explained by the very low autoxidation rate for AtHb1 compared to the latter ones [64]. This observation may partly explain why the reaction rates are higher in acidic pH: during autoxidation, superoxide anion formation is favored in lower pH, which could, in turn, generate hydrogen peroxide, thus leading to ferryl intermediate formation. 

### 2.4. EPR Measurements in Phytoglobins Reactivity towards Nitrite

Electron paramagnetic resonance spectra were also measured to gain a better understanding of whether the species formed as either intermediates or as final products. Parameters extracted from EPR spectra such as g factors and hyperfine coupling constants can help to identify the paramagnetic species involved, as well as their spin state, geometry, and coordination sphere approximate composition, while characteristics such as linewidth values offer valuable insights on how paramagnetic centers interact with one another (spin–spin coupling interaction). 

Metglobins (Fe^3+^) EPR spectra were measured for each protein discussed in this paper in both the presence and absence of nitrite, as shown in Figure 6. Both metAtHb1 and metAtHb2 appear to be EPR silent at 100 K. Here, temperature plays a central role; as both nonsymbiotic hemoglobins are hexacoordinated, their EPR spectrum should result from the binding of histidine (through one of the nitrogen atoms) to the Fe^3+^ center, resulting in a low-spin heme adduct. Such centers are known to exhibit very short relaxation times with wider signals due to the g-strain effect. As such, the lack of EPR signal for both met forms of the nonsymbiotic hemoglobins is due to faster relaxation at 100 K rather than the lack of a paramagnetic center. As both metAtHb3 and metMb are pentacoordinated, their EPR spectrum shows an intense signal at g factors equal to 5.9, thus indicating a high-spin adduct, most probably due to an aqua molecule bound in the sixth heme coordination position. However, when nitrite is added to met forms, the resulting spectra become more interesting. Upon nitrite addition, AtHb2 remains EPR silent, while myoglobin, based on intensity difference, shifts to become a MetMb–NO_2_^−^ adduct, which appears to be a rhombic low-spin signal of a paramagnetic species that relaxes very fast, as described in [67]. MetAtHb1 generates a small signal, based on which a presumed rhombic geometry, a g_yy_ factor different from two, and a width spanning around 1000 Gauss can be assigned to metAtHb1-NO_2_^−^. Even if both nonsymbiotic phytoglobins are hexacoordinated, those in class 2 have a higher affinity for distal histidines, thus explaining why one of them may form an adduct with nitrite while the other does not [68]. What is certain is that neither of these three globins exhibits any kind of reactivity towards nitrite in their met state. However, a closer examination of the EPR spectrum resulting from the interaction between metAtHb3 and nitrite shows an unexpected signal that is not characteristic of a low-spin Fe^3+^-NO_2_^−^ adduct. The absence of any high-spin signals rules out the possibility of the formation of a high-spin Fe^3+^-NO_2_^−^ adduct as well. Moreover, when compared to deoxyAtHb3 treated with nitrite, both spectra show similar features with a g factor of two; this may indicate that both metAtHb3 and deoxyAtHb3 can react with nitrite to generate an intermediate or a final product characterized by a free radical with a metallic component (thus the higher width of the signal). The fact that metAtHb3 could exhibit any reactivity towards nitrite might be explained by its unique binding site architecture, which consists of tyrosine and tryptophan, both redox-active amino acids. However, variable low-temperature measurements are needed to check whether or not such an activity indeed exists. When treated with nitrite, the EPR spectrum of oxyAtHb3 is EPR silent.

As shown in Section 2.2, the studied ferrous (deoxy) phytoglobins exhibit nitrite reductase-like activity to varying degrees. Nitrosyl–hemoglobin adducts have a characteristic EPR signal due to their free radical character and, as shown in Figure 7, all three deoxyphytoglobins generate such species after 1 min incubation with nitrite. These signals indeed exhibit a g factor closer to that of the free electron, thus indicating a free radical character. The width of the signal also matches previously described nitrosyl–hemoglobin adducts. Isotopic exchange based on reacting nitrite with either ^14^N (S = 1) or ^15^N (S = 1/2) indicates that parts of the hyperfine splittings arise from the interaction between Fe^2+^ and a NO adduct. Slight variations in hyperfine features can be easily attributed to binding pocket particularities, as described in Section 2.1. DeoxyMb seems to form an EPR-silent adduct, far from the distinguishable nitrosylMb adduct shown as a control in the lower half of Figure 7 and generated using nitric oxide dissolved in buffer solution. It is still interesting to note that, while the nitrosylMb signal is intense and has a clear free radical character, its hyperfine components are hardly resolved [69].

### 2.5. Mechanistic Considerations in Phytoglobins Reactivity towards Nitrite

The nitrite reductase activity of nonsymbiotic class 1 and class 2 hemoglobins from *Arabidopsis thaliana* has been previously reported. It was observed that both deoxyAtHb1 and deoxyAtHb2 generate NO gas and nitrosylhemoglobin species [40]. We observed nitrosylHb formation through EPR spectroscopy, as described in Section 2.4. DeoxyAtHb1 shows, by far, the better conversion rate of the two, an aspect that can be explained by its lower affinity for distal histidine. A reassessment of reaction kinetics and stoichiometry of the reaction between nitrite and human deoxyHb was reported, highlighting that the process is more complex than had been previously implied [70]. Under strictly controlled anaerobic conditions and in the absence of any reducing agent, the final product of such a reaction with human hemoglobin is an equimolar mixture of metHb and nitrosylHb. What is interesting is that the kinetic curve is a sigmoidal one, suggesting allosteric control, with an apparent rate of 0.8 M^−1^s^−1^. Neither class 1 nor class 2 phytoglobins exhibit such a kinetic profile, which is explained by their higher rates, as seen in Table 1. The met form was not detected for either phytoglobin due to the use of dithionite as an electron source. It also turned out that, at least for human deoxyhemoglobin, the R and T states have different reactivities towards nitrite, with the former accelerating the reaction. It was previously shown that the mutagenesis of myoglobin amino acids from the binding pocket greatly affects the nitrite reductase-like reactivity. Based on this, the difference in reactivity between the four globins studied in this paper stems from the architecture of the secondary coordination sphere surrounding the heme, which controls the nitrite binding and fine-tunes the pKa.

When it comes to the mechanism, two models may explain the observed nitrite reductase activity: (i) a nitrate reductase model in which nitrite and deoxyHb interact and generate metHb and NO, which subsequently binds to deoxyHb, and (ii) a nitrite anhydrase model which implies that nitrite is first converted to N_2_O_3_ and then to NO and NO_2_, with the final products being, yet again, metHb and nitrosylHb [70]. A very general mechanistic overview can be seen in Figure 8. At higher nitrite concentrations relative to deoxyHb, both models fit the observed kinetics, at least for human hemoglobin. None of the nonsymbiotic hemoglobins exhibited met features in either their UV-vis profiles or EPR spectra in our experiments, as suggested by the nitrite reductase mechanism, while the formation of nitrosylphytoglobin adducts was obvious, as shown in Figure 7. However, in our protocol, excess dithionite was used, and thus, all generated metglobins were reduced back to deoxy form. The lack of a nitrosylMb adduct detected by EPR spectroscopy may be correlated with deoxyMb reduced reactivity towards nitrite, as the incubation time was relatively short (approximately 1 min). 

The reaction of nitrite with oxyhemoglobin (oxyHb) generates nitrate and methemoglobin (metHb); the mechanism is a very complex one that is presumed to involve protein-generated free radicals, hydrogen peroxide, nitric dioxide, and ferrylglobins as intermediates [64]. The kinetic curve that describes this reaction, shown in Figure 5 and Appendix A, can be broken down into two main phases: the lag phase, where presumably a reactive species accumulates, and a more rapid phase characterized by abrupt absorbance decay, which resembles a radical-type propagation phase. Previous mechanistic investigations established that when catalase is added at the reaction onset, the rate is far slower and the lag phase is longer, thus indicating that hydrogen peroxide is not only generated but also plays an important part in driving the process toward the propagation phase; the addition of catalase after the lag phase does not affect the reaction rate, meaning that, indeed, hydrogen peroxide is crucial to initiating the oxidant, which, based on the pH dependence, is most probably a ferrylglobin [64]. It is interesting to note that, as shown in Figure 6, when oxyAtHb3 was treated with nitrite, a ferryl intermediate can be inferred from the presence of a free radical signal in the EPR spectrum. As for our previous experiments, ferrylAtHb3 was the most reactive Fe(IV) = O species among all four studied globins due to its unique binding site architecture, where both tyrosine and tryptophan can be found, and recording its spectrum in the presence of a substrate was very difficult. This can also be seen in the kinetic data associated with Appendix A, where it is clear that oxyAtHb3 is the fastest at converting nitrite to nitrate. Taken together, all these observations suggest that a ferryl intermediate is involved, as was previously observed by the use of stopped-flow spectrophotometry [71], at least when phytoglobins are used as catalysts. If this ferrylphytoglobin is produced by either a direct interaction with hydrogen peroxide or from peroxynitrate intermediate decomposition remains to be seen. DFT calculations revealed an essentially barrierless reaction between nitrite and oxyMb, with a notable outer-sphere component; a metastable ferrous–peroxynitrate adduct was found to feature a very low barrier towards nitrate liberation, with ferryl as a final product [71]. However, ferryl production with hydrogen peroxide would require the existence of a metglobin somewhere in the mechanism. It is also important to note that superoxide dismutase does not affect the reaction at any phase, which means that superoxide is not involved at all. Both possible pathways of ferrylglobin generation are shown in Figure 8. 

This two-faced reactivity towards nitrite may account for more than just a response to hypoxia. The reactivity of oxyphytoglobins towards nitrate was previously explored *in vivo* and may form an alternative respiratory pathway during hypoxia, as NO_3_^−^ plays the role of an intermediate electron acceptor. It was shown that nitrate supplementation improves hypoxia tolerance, but data regarding uptake during low-oxygen conditions are contradictory [72,73,74,75]. Nitrite would be the product of nitrate reduction, but it would not accumulate in plants, as it will be subsequently reduced to nitric oxide in hypoxia. NO was linked to the regulation of ethylene biosynthesis, which, in turn, controls adaptation to hypoxia. The reduction of nitrite can be coupled with proton translocation to maintain ATP biosynthesis [7]. What is even more interesting is that, to ensure a form of anaerobic respiration, other enzymes, presumably oxyphytoglobins, based on the mechanism discussed in this paper, can oxidize NO back to nitrate so that the cycle can continue [76]. The scavenging of nitric oxide is essential, as its accumulation can lead to cell death [77]. While class 1 phytoglobins are characteristic of mostly hypoxic roots and germinating seedlings, class 2 are non-hypoxia-inducible but expressed under normoxic conditions in the roots and shoots of adult plants. Interestingly enough, class 1 phytoglobins were also detected in non-stressed leaves, shoots, and inflorescences. High levels of nitric oxide are mostly associated with hypoxia, but the basal distribution of NO is connected to normal growth in *A. thaliana*; it was shown that class 1 and class 2 phytoglobins are responsible for controlling the levels of NO in stem cells, thus triggering organ differentiation [78]. Class 3 phytoglobins are mostly down-regulated in hypoxia and are found in higher concentrations in roots as compared to stems. However, their function under normoxia remains elusive [36]. 

## 3. Materials and Methods

### 3.1. Cloning and Expression of the Recombinant nsHs Phytoglobins

The three nsHbs were cloned from *A. thaliana* cDNA as we previously described [49]. Briefly, after construct amplification, successive BP and LR reactions followed by cloning the constructs into the Gateway pVP16 expression vector that contains the coding sequence for N-terminal 8xHis:MBP double affinity tag flanked by a TEV recognition sequence. Following the heat shock transformation of the expression vectors into competent BL21(DE3) *Escherichia coli* cells, the recombinant nsHbs were obtained in great yields. The Hbs were fused with MBP, which is an affinity tag that can be used to facilitate the purification of the protein of interest. The most important reason for whey we chose MBP as a fusion partner rather than some other affinity tag is its remarkable ability to enhance the solubility of its fusion partners [79]. Transformed *E. coli* cells containing the expression vector were grown at 37 °C and 190 rpm in Luria–Bertani (LB) medium and 100 mg/L ampicillin until the optical density at 600 nm reached about 0.6 for AtHb1 and AtHb2, after which the temperature was reduced to 25 °C, and protein expression was induced by 150 μM isopropyl β-d-1-thiogalactopyranoside (IPTG) and 50 μM heme that had been dissolved in slightly alkalized water. The cells were further grown at 25 °C for at least eight hours or overnight.

In the case of AtHb3, the cells were grown at 37 °C and 190 rpm in Luria–Bertani (LB) medium supplemented with 100 mg/L ampicillin until the absorption at 600 nm reached about 1.2, after which the temperature was reduced to 28 °C, and protein induction was carried out with 0.3 mM IPTG, 0.1 mM ferrous ammonium sulfate, and 0.25 mM 5-aminolevulinic acid as heme precursors. Replacing the heme with a heme precursor leads to higher yields of much cleaner hemoglobin and prevents heme-generated free radicals during cell lysis. Additionally, a 40 mL sterilized LB medium saturated in CO, previously prepared by purging the gas directly into the LB medium for approximately 20 min, was added before the flask was sealed. The cells were incubated overnight at 28 °C, and the shaking was reduced to 110 rpm. The cells were harvested by centrifugation at 4000 rpm for 20 min at 4 °C and suspended in a 100 mL lysis buffer pH 8 containing 300 mM NaCl and 50 mM Na_2_HPO_4_, which was previously bubbled with CO gas for 15 min, in the case of AtHb3 only. The CO gas further prevents the oxidative oligomerization of the AtHb3 hemoglobin. The cells were sonicated for 15 min on ice in the presence of 1 mM phenylmethanesulfonyl fluoride (PMSF). Cellular debris was centrifuged at 16,000 rpm for 45 min at 4 °C, and a bright red supernatant containing the soluble proteins was collected for further purification. Purification was carried out in three steps.

### 3.2. Purification of the Recombinant nsHs Plant Hemoglobins

The first rough purification was carried out using Ni–NTA agarose affinity columns (GE Healthcare Life Sci, Hino, Japan) according to the manufacturer’s manual with slight modifications. The fractions of carboxyAtHb3 (Hb-Fe^2+^-CO), oxyAtAb1, and oxyAtHb2 that bind to the Ni-NTA resin were firstly oxidized to ferric form (Hb-Fe^3+^) directly on the column before elution using 10 mM solution of potassium ferricyanide (K_3_Fe(CN)_6_) and then washed at least four times the column volume with 300 mM NaCl, 50 mM Na_2_HPO_4_ buffer, pH 7.8; elution was performed using the same buffer containing 200 mM imidazole. The second purification was carried out using FPLC (ÄKTA purifier, Cytiva, Tokyo, Japan) on 5 mL MBPTrap HP columns (GE Healthcare Life Sci) using 150 mM NaCl, 50 mM Na_2_HPO_4_, pH 7.8 as a binding buffer; elution was performed with 10 mM maltose dissolved in binding buffer. The last purification step comprised size-exclusion chromatography, which was carried out with a HiPrep 26/60 Sephacryl S-200 HR column (GE Healthcare Life Sci) using 50 mM Na_2_HPO_4_, 150 mM NaCl buffer, pH 7.8 as a running buffer.

To cleave the affinity tag, fusion recombinant Hbs were treated with 27 kDa Tobacco Etch Virus (TEV) protease in a ratio of A280 (TEV)/A280 protein = 1/50 in the presence of 3 mM glutathione and 0.5 mM EDTA overnight at 4 °C. The cleaved fractions were separated using a system of 5 mL HisTrap HP connected to 3 × 1 mL MBPTrap HP columns (GE Healthcare Life Sci) using 300 mM NaCl, 50 mM Na_2_HPO_4_, pH 8 as a binding buffer; the cleaved AtHbs were collected in the flow-through.

The Hbs were purified to homogeneity and assessed by 15% SDS-PAGE electrophoresis with samples before and after TEV protease cleavage as previously shown [49] and as shown in Appendix A.

### 3.3. Preparation of Oxy- and Deoxyglobins 

Horse heart metmyoglobin (metMb) was prepared by dissolving the lyophilized protein (Sigma, St. Louis, MO, USA) in 50 mM sodium phosphate (pH 7.4) and roughly desalted on the Sephadex G-25 column (Cytiva) equilibrated in the same buffer. Deoxyglobins and deoxyMb (GlbFe^2+^) were prepared *in situ* in 50 mM phosphate buffer (pH 7) at 25 °C in an anaerobic cuvette by adding sodium dithionite until full reduction was achieved; excess dithionite was monitored at 315 nm. Oxyglobins (GlbFe^2+^-O_2_) were prepared by desalting freshly prepared deoxyglobins on PD10 containing a Sephadex G-25 resin (GE Healthcare Life Sci.) using a 50 mM sodium phosphate buffer (pH 7.4). The quantification of all protein forms was performed using the extinction coefficients that we previously determined [49].

### 3.4. UV-Vis Spectrophotometric Measurements

Spectrophotometric data were acquired using a Varian Cary^®^ 50 UV-Vis Spectrophotometer (Agilent, Santa Clara, CA, USA) equipped with a temperature-controlled multi-cell holder using a 1 cm anaerobic quartz cuvette for the anaerobic measurements and the Microplate Reader Tecan Infinite M1000 (Tecan Systems, Kawasaki, Japan) for the aerobic experiments.

#### 3.4.1. The Reaction of Deoxyglobins (GlbFe^2+^) with Nitrite 

The reaction kinetics of nitrite reduction by deoxyferrous forms (HbFe^2+^) was studied under anaerobic conditions using 8 μM of each deoxyglobin (AtHb1, AtHb2, AtHb3, and Mb) mixed with 0.05–1 Mm NaNO_2_ (from a deoxygenated stock solution, transferred with a gastight Hamilton syringe), in the presence of 5 mM dithionite in pH 7 phosphate buffer (50 mM) at 25 °C. UV-vis spectra were continuously acquired between 300 and 700 nm. The traces extracted from the Soret band (425 nm for both AtHb1 and AtHb2, 432 nm for AtHb3, and 435 nm for Mb) for each reaction at different nitrite concentrations were fitted in a single-exponential manner, and *k_obs_* (min^−1^) values were calculated. The values of *k_obs_* (min^−1^) against nitrite concentration were plotted, and the rate constants for the bimolecular reaction of deoxyglobins (GlbFe^2+^) with nitrite were calculated. 

#### 3.4.2. The Reaction of Oxyhemoglobins (GlbFe^2+^-O_2_) with Nitrite 

Aerobic measurements of nitrite reduction by freshly prepared oxyglobin forms (HbFe^2+^-O_2_) were performed in controlled aerobic conditions. A total of 8 μM oxyglobin was mixed with appropriate amounts of stock solution of NaNO_2_ at the specific concentrations indicated in the text/figures. Reaction progress was monitored by observing the decrease in the amount of oxyferrous heme at 575 nm for AtHb1 and AtHb2, 578 nm for AtHb3, and 582 nm for Mb. All the experiments were performed in 50 mM phosphate buffer at pH 7 and 25 °C. The reaction profile was further investigated at various pHs in the range of pH 6–9.5.

### 3.5. EPR Measurements

The EPR samples were prepared by incubating 200 μM of each deoxyglobin with 5–10 mM of either Na^14^NO_2_ or Na^15^NO_2_ in pH 7 phosphate buffer (50 mM) for one minute. Each mixture was then transferred to an EPR quartz tube and frozen in liquid nitrogen. The EPR spectra were measured at 100 K in CW mode using an X-band MicroEMX spectrometer (Bruker, Billerica, MA, USA) equipped with a liquid nitrogen cooling system. Each spectrum was initially measured over 4000 Gauss, with a center field at 2500 Gauss, 10 mW microwave power, 40.96 ms time constant, and a modulation amplitude of 3 Gauss. Some samples were also measured over narrower fields, with a center field equal to 3300 Gauss and a sweep width of 600 Gauss. Measurements with three scans were performed for each sample.

## 4. Conclusions

Despite their structural and possible functional differences, all three nonsymbiotic plant hemoglobins from *A. thaliana* display a nitrite reductase-like activity under anaerobic conditions, along with better rate constants than horse heart myoglobin. The highest rate constant is that of AtHb1, 67.56 M^−1^s^−1^, which may also act as an in vivo nitrite reductase. Nitrosylglobin EPR signals were detected for all three phytoglobins upon the treatment of their deoxy forms with nitrite. All the studied globins reacted with nitrite in aerobic conditions as well; among them, AtHb3 is by far the most reactive one with regard to this function. Ferryl implication was also discussed based on spectroscopic and kinetic observations. Together, the data show that phytoglobins indeed interact with nitrite in both anaerobic and aerobic conditions and have better kinetic parameters compared to mammalian globins. The current study’s results may point to *in vivo* phytoglobins’ involvement in the nitrate–nitrite–NO anaerobic respiration pathway. 

## Figures and Tables

**Figure 1 molecules-29-01200-f001:**
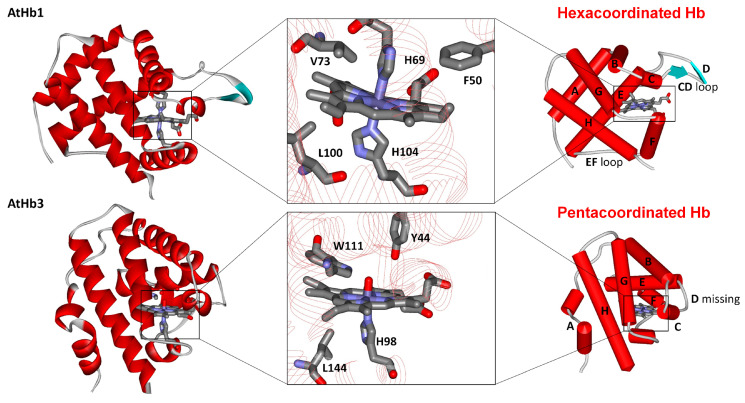
The two representative folding structures for nonsymbiotic phytoglobins from *Arabidopsis thaliana* (AtHb1, 3ZHW, pdb code) and the truncated one from the same organism (AtHb3, 4C0N, pdb code). The coordination differences are strikingly visible. While AtHb1 is hexacoordinated and has a 3/3 fold (helices A, E, F on B, G, H), AtHb3 is pentacoordinate and has a 2/2 fold (helices B, E on G, H). Figure reproduced here with permission from [48].

**Figure 2 molecules-29-01200-f002:**
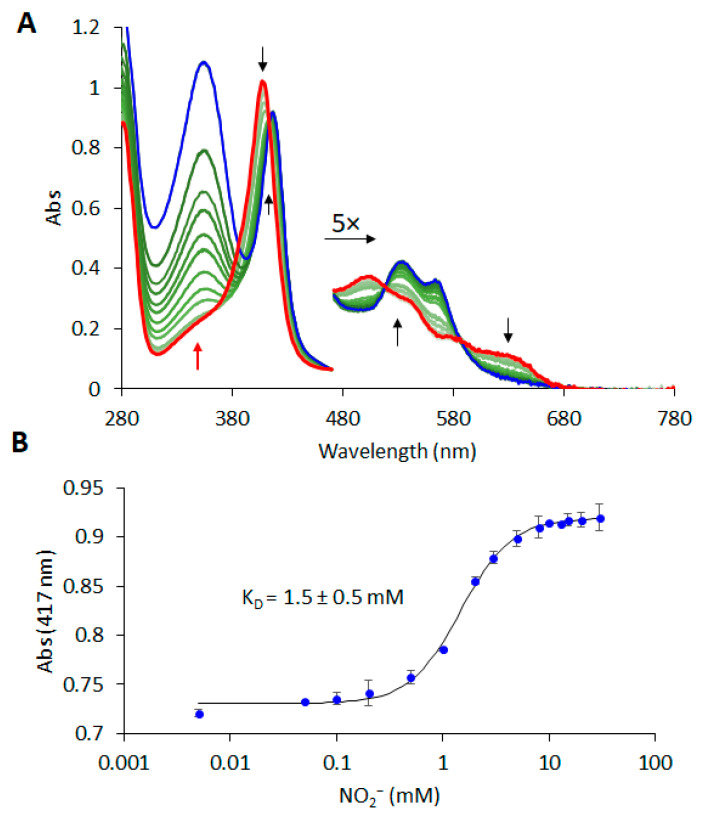
**UV-vis spectral changes for the titration of recombinant AtHb3 methemoglobin (HbFe^3+^) with nitrite in aerobic conditions.** UV-vis spectra of 8 μM methemoglobin (HbFe^3+^) after the addition of nitrite at different concentrations from 5 μM to 30 mM in 50 mM phosphate buffer at pH 7 and 25 °C. (**A**) Spectral monitoring for the full 280–780 nm domain. The vertical black arrows indicate the directions of the spectral features that exhibit striking changes, and the upward red arrow at 360 nm indicates the nitrite band that increases in absorbance as its concentration rises. The 480–780 nm domain was multiplied with a factor of 5 for better visibility. The initial spectrum (without nitrite) is colored in red, and the final spectrum (at 30 mM nitrite) is colored in blue, whereas at intermediate nitrite concentrations, the spectra are colored in hues of green, from light green (low concentration of nitrite) to darker green (high concentration of nitrite). (**B**) Titration curve at 417 nm (specific for AtHb3Fe^3+^-NO_2_^−^ adduct) and the value for K_D_ as determined by sigmoidal fitting for log concentration axis.

**Figure 3 molecules-29-01200-f003:**
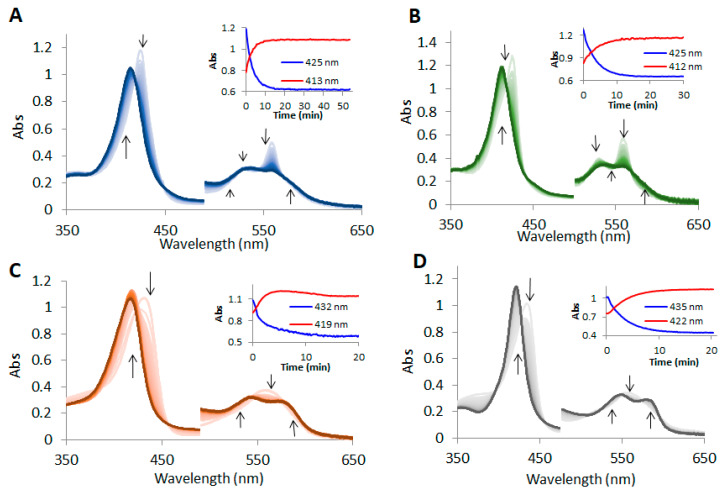
**The reaction between nitrite and recombinant *Arabidopsis thaliana* nonsymbiotic deoxyhemoglobins (HbFe^2+^) and horse heart deoxyMb.** UV-vis spectra of 8 μM deoxyhemoglobins (HbFe^2+^) during oxidation by nitrite in 50 mM phosphate buffer at pH 7 and 25 °C; (**A**) 8 μM deoxyAtHb1 with 0.05 mM nitrite; (**B**) 8 μM deoxyAtHb2 with 0.1 mM nitrite; (**C**) 8 μM deoxyAtHb3 with 0.25 mM nitrite; and (**D**) 8 μM deoxyMb with 0.5 mM nitrite. In all cases, 5 mM dithionite was used to ensure an anaerobic medium and the reduction of the formed ferric forms. The specific monochromatic color for each form of Hb is transient from a light hue to a dark hue. The 480–780 nm domain was multiplied with a factor of 5 for better visibility. The vertical black arrows indicate the directions of the spectral features that exhibit striking changes. Insets: the kinetic profile of the reactant consumption (blue) and the kinetic profile of the product formation (red) are shown.

**Figure 4 molecules-29-01200-f004:**
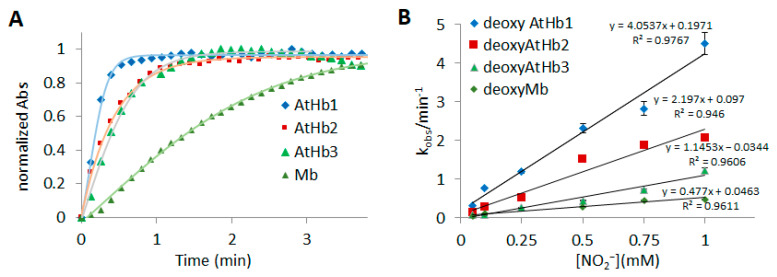
**Kinetics of nitrite reaction with *Arabidopsis thaliana* nonsymbiotic deoxyhemoglobins (HbFe^2+^) and horse heart deoxyMb**. (**A**) Time course of the reaction of each Hb and Mb at 1 mM nitrite. The absorbance change is normalized to that associated with the transition from each deoxyhemoglobin (413 nm for AtHb1, 412 nm for AtHb2, 419 nm for AtHb3, and 422 for Mb) to its endpoint spectrum. (**B**) Plot of the observed rate constants (*k_obs_*/min^−1^) versus nitrite concentration obtained in 50 mM phosphate buffer at pH 7 and 25 °C.

**Figure 5 molecules-29-01200-f005:**
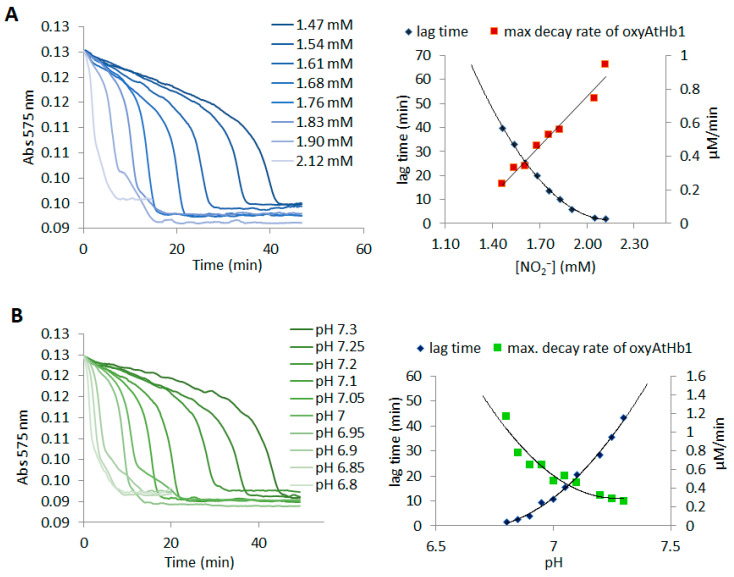
**Kinetic profiles and their corresponding lag times and maximum decay rates for the reaction between oxyAtHb1 and nitrite.** (**A**) At different concentrations of sodium nitrite, 8 μM oxyAtHb1 was incubated with NaNO_2_ (1.47–2.12 mM) in 50 mM phosphate buffer (pH 7). (**B**) At different pH values, 8 μM oxyAtHb1 was incubated with 1.83 mM NaNO_2_ in a buffer with pH 6.8–7.3.

**Figure 6 molecules-29-01200-f006:**
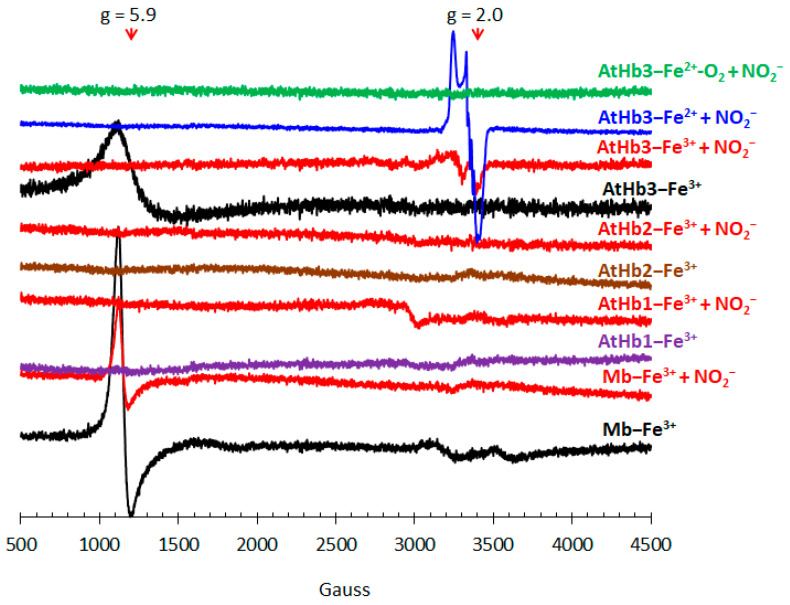
**EPR spectra of the studied met form of the Hbs (200 μM) before and after cca. 1 min of NaNO_2_ treatment.** The spectra were measured in 50 mM phosphate buffer (pH 7, temperature 100 K, X-band), and the cavity spectrum was subtracted from all spectra. For comparison, in the case of AtHb3, the deoxy (**in blue**) and oxy (**in green**) forms were also included. The spectrum of the deoxy form after nitrite treatment was divided by 2 so that it better fits the scale.

**Figure 7 molecules-29-01200-f007:**
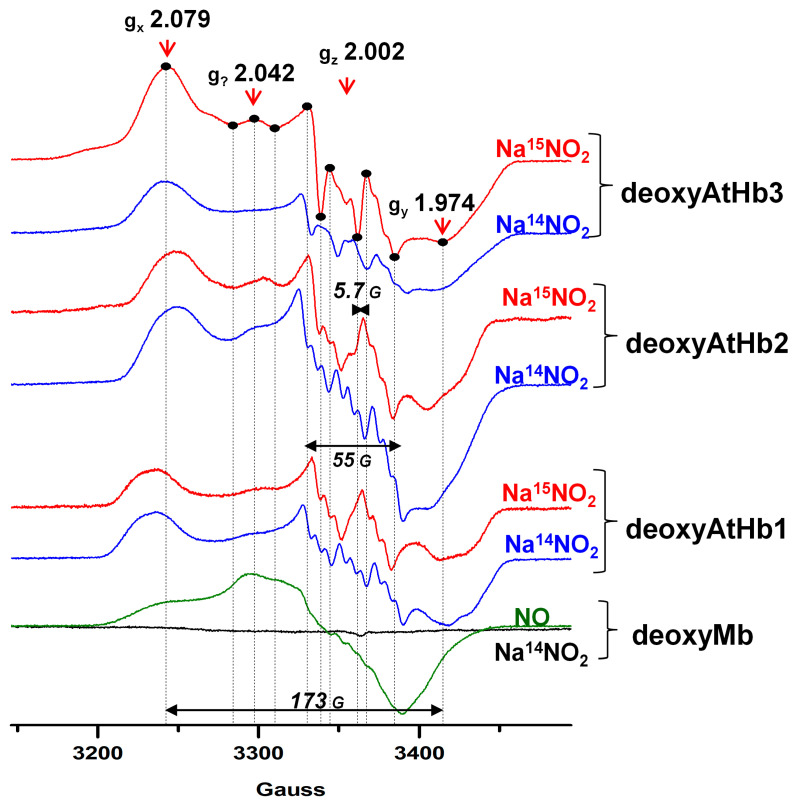
**EPR spectra of the deoxyHbs (200 μM) after cca. 1 min of NaNO_2_ treatment in 50 mM phosphate buffer (pH 7, temperature 100 K, X-band).** The g values and coupling constants are indicated. Note that Mb shows no detectable features, and the expected ferrous Mb nitrosyl is presented as a reference (**green**). Black dots in the top spectrum have no other meaning except for the identification of the gridlines added for the x-axis.

**Figure 8 molecules-29-01200-f008:**
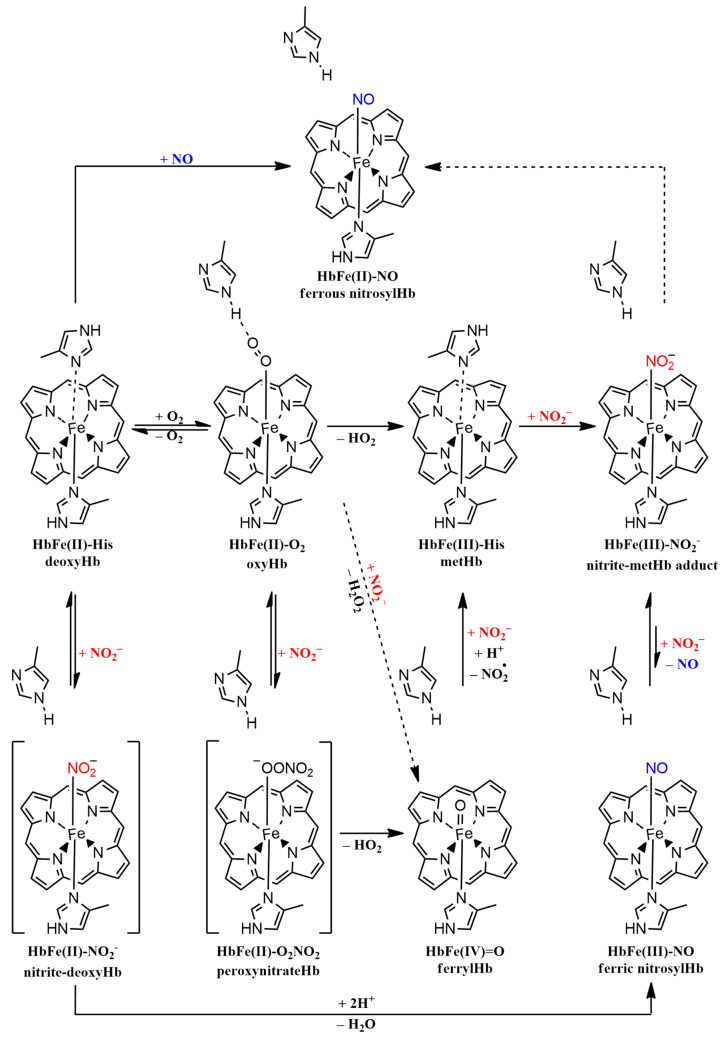
Tentative mechanism illustrating both nitrite reductase activity for deoxyHbs and nitrite oxidase activity for oxyHbs. The dashed arrows indicate the more complex reactions that take place in several steps, with potential reaction intermediates not being depicted here.

**Table 1 molecules-29-01200-t001:** Rate constants for the bimolecular reaction (M^−1^s^−1^) of deoxyHbs (HbFe^2+^) with nitrite.

AtHb1	AtHb2	AtHb3	Mb
67.56	36.62	19.09	7.95

## Data Availability

The data presented in this study are available upon request from the corresponding author.

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
