# Peer review of "Redox Reactivity of Nonsymbiotic Phytoglobins towards Nitrite"

_molecules, 2024, doi:10.3390/molecules29061200_

Round 1
Reviewer 1 Report
Comments and Suggestions for Authors
Zagrean-Tuza et al. describe the reactivity of nonsymbiotic phytoglobins from Arabidopsis thaliana (AtHb1-3) towards nitrite. The ability of AtHb to act as a real nitrite reductase is discussed. The ability of AtHb1 and AtHb2 was already studied by the same experimental approach (Tiso et al. Biochem, 2012), so the novelty relays only for AtHb3. In my opinion, the biological implications of this activity in plants are not well discussed. Some Figures and legends should be improved. Many experiments are not shown and, in my opinion, should be presented as supplemental figures.
Line 134: “The binding of the nitrite to the met form could not be detected using UV-vis spectroscopy up to tens of millimolar range, probably due to the coordination of the distal histidine.” It should be shown as a Supplemental Figure.
Fig. 2. A better explanation, as well as a description of Figure legends, is needed. Is this performed in aerobic conditions? What are the blue and red curves? What are the green curves? The peak at 350 nm is not described. Please detail
Fig. 3. Which is the difference between Fig 2 and 3? Please add a reference for the insets and the colors of the different curves observed.
Figure 4. Please expand the explanation. Which absorbance is monitored in Fig. 4A?
Line 176: “In view of the pH used in these experiments, these results are in good agreement with previous works”, please detail the pH used. Compare data with the previous works.
Line 170: These findings support the previous observations that nitrite reduction leads to nitric oxide and ferric Hb 171 is reduced by dithionite. Which previous observation? Please cite reference
In my opinion, Figure 5,6,7 should be condensed into one figure (with some supplemental data) to compare kinetics easily. Figure 8 is not mentioned in the text. I suggest moving it to supplemental data.
As I am not a specialist in EPR is rather difficult to follow this experiment. I should be able to explain it to a broader audience.
Figure 11. What are the dashed arrows? Please complete the Figure legend.
A measurement of NO emission should be performed to confirm the author´s hypothesis.
A complete discussion about the physiological context in which AtHbs (1-3) is expressed in plants (also at subcellular localization) and may act as an in vivo nitrite reductase is needed.
Line 97: Why there is an increase in nitrite during hypoxia? (please provide a reference) Is Hb3 induced under hypoxia?
The nitrite reductase activity observed for all three Hbs is always under anoxia? In which other physiological conditions could appear?
Reviewer 2 Report
Comments and Suggestions for Authors
The authors are encouraged to submit a new version of the present manuscript. There are some serious flaws that need to be handled before publication:
The titile of the manuscript appears not to be fully relevant. The comparison with myoglobin seems inappropriate. When examining the Abstract section, nothing is even mentioned about myoglobin. The authors mention mammalian globins, but these are indeed different.
The handling of the reference list seems very sloppy. References are missing and sometimes used in the wrong context in the manuscript. Some journal names are abbreviated, others not, some journal names are missing.
The methods used for isolation of the phytoglobins used are very different. The authors need to dwell on this much more. Different methods are thus used for the three phytoglobins presented and the authors need to discuss why they have chosen these alternative approaches. Variations in the production phase may influence folding and heme binding. Nothing is said about their purities.
The interpretation of the kinetic reactions are not fully obvious. It would be interesting to compare the figures in more depth. How can the diffreences be explained? Is only a single reaction involved or are two reactions overlapped.
Comments on the Quality of English LanguageTheir are only some minor points that need to be corrected,
Reviewer 3 Report
Comments and Suggestions for Authors
In the reviewed study, three representative nonsymbiotic phytoglobins (nsHbs) of the plant model Arabidopsis thaliana were presented and their nitrite reductase like activity and involvement in nitrosative stress were discussed. The reaction kinetics and mechanism of nitrite reduction by nsHbs (deoxy and oxy form) at different pHs were investigated by means of UV-Vis spectrophotometry, along with EPR spectroscopy. The reduction of nitrite requires an electron supply and it is favored in acidic conditions. This reaction is critically affected by molecular oxygen, since oxyAtHb will catalyze nitric oxide deoxygenation.
In my opinion, the paper is properly designed and written. It is quite interesting, however, I formulated some suggestions/recommendations:
- The Introduction is too long and overloaded in detailed information, therefore, it should be revised into a more concise form,
- Due to the inclusion of numerous figures in the article, please consider moving some of them to the Supplementary file,
- In the Materials and Methods section, the Authors stated that “Hbs were assessed by 15% SDS-PAGE electrophoresis with samples before and after” – hence, I recommend to place these electropherograms (e.g. TIFF files) in the Supplementary file,
- Moderate editing of English language is recommended.
Comments on the Quality of English LanguageModerate editing of English language is recommended.
Round 2
Reviewer 1 Report
Comments and Suggestions for Authors
The authors have responded to all my observations and modified the article accordingly. However, I recommend the authors revise the entire manuscript to fix any errors in English grammar.
Comments on the Quality of English Language
I recommend the authors revise the entire manuscript to fix any errors in English grammar and misspelling.
Line 111: “in the presence of NADH”
Line 131: “while an increase”
Line 139: “slightly better than that of myoglobin”
Line 274: “Figure 5”
Line 409: “The met form was not detected for phytoglobin due to the use of dithionite as electron source.”
Line 414: “Based on this, the difference in reactivity between the four globins studied in this paper stems from the architecture of the secondary coordination sphere surrounding the heme, which controls the nitrite binding and fine-tunes the pKas”
Line 435: “Figure 8” instead of 11
Line 449: “is crucial to initiating the oxidant”
Line 485: “in non-stressed leaves”
Line 490: “are found in higher concentrations”
Line 612: Please rewrite this sentence. “More exploration is needed in order to fully grasp the complete mechanism behind this activity in order to integrate these processes as plant physiological important.”
Author Response
We thank the reviewer and academic editor for their suggestions, we implemented all of them and beyond and we answer all the raised comments bellow. We consider that the second revision of the manuscript furthered improved its quality.
Reviewer 1
Comments and Suggestions for Authors
The authors have responded to all my observations and modified the article accordingly. However, I recommend the authors revise the entire manuscript to fix any errors in English grammar.
Comments on the Quality of English Language
I recommend the authors revise the entire manuscript to fix any errors in English grammar and misspelling.
Line 111: “in the presence of NADH”
Line 131: “while an increase”
Line 139: “slightly better than that of myoglobin”
Line 274: “Figure 5”
Line 409: “The met form was not detected for phytoglobin due to the use of dithionite as electron source.”
Line 414: “Based on this, the difference in reactivity between the four globins studied in this paper stems from the architecture of the secondary coordination sphere surrounding the heme, which controls the nitrite binding and fine-tunes the pKas”
Line 435: “Figure 8” instead of 11
Line 449: “is crucial to initiating the oxidant”
Line 485: “in non-stressed leaves”
Line 490: “are found in higher concentrations”
Reply: We thank the reviewer for pointing out our grammar and misspelling errors. We fixed everything mentioned by the reviewer and more. Please see all our changes pertaining to grammar, spelling and phrasing in the reviewed version of the manuscript (track changes activated).
Line 612: Please rewrite this sentence. “More exploration is needed in order to fully grasp the complete mechanism behind this activity in order to integrate these processes as plant physiological important.”
Reply: We rephrased the last part of our Conclusions section: The current results may point to in vivo phytoglobins involvement in the nitrate-nitrite-NO anaerobic respiration pathway.
Academic Editor Notes
Accept after minor revision. Although I strongly appreciate the work done by the authors to improve the manuscript, I agree with Reviewer 1 that some errors in English grammar and misspelling need to be fixed. Moreover, references 1, 18, 23 and 48 should completed and/or corrected
Reply: We fixed our English language errors to the best of our ability (please see the document with the track changes activated). We corrected the references mentioned.